# Quercetin Induces Anticancer Activity by Upregulating Pro-NAG-1/GDF15 in Differentiated Thyroid Cancer Cells

**DOI:** 10.3390/cancers13123022

**Published:** 2021-06-16

**Authors:** Yukyung Hong, Jaehak Lee, Hyunjin Moon, Chang H. Ryu, Jungirl Seok, Yuh-Seog Jung, Junsun Ryu, Seung J. Baek

**Affiliations:** 1College of Veterinary Medicine and Research Institute for Veterinary Science, Seoul National University, Seoul 08826, Korea; hyu1154@naver.com (Y.H.); ljh930307@snu.ac.kr (J.L.); hyjin1995@naver.com (H.M.); 2Center for Thyroid Cancer, Department of Otolaryngology-Head and Neck Surgery Research Institute and Hospital, National Cancer Center, Goyang-si 10408, Korea; changhwanr@ncc.re.kr (C.H.R.); jseok@ncc.re.kr (J.S.); jysorl@ncc.re.kr (Y.-S.J.)

**Keywords:** thyroid cancer, quercetin, pro-NAG-1, GDF15, C/EBP, mature NAG-1

## Abstract

**Simple Summary:**

Thyroid cancer is one of the most common cancers worldwide, and its incidence has increased over the last few decades. It is difficult to diagnose different types of thyroid cancer. Tumor tissues from papillary thyroid cancer patient showed higher expression of mature NAG-1, whereas adjacent normal tissues showed higher expression of pro-NAG-1. Several anti-cancer compounds increased pro-NAG-1 expression in thyroid cancer cell line. Quercetin (3,3’,4’,5,7-pentahydroxyflavone) is a flavonoid that is a major component of various plants, including raspberries, grapes, and onions. Quercetin induced apoptosis by inducing only pro-NAG-1 expression, but not mature NAG-1, mediated by the transcription factor C/EBP. This study indicates that pro-NAG-1 could be used as a useful biomarker for thyroid cancer and also provides a potential therapeutic target for the treatment of thyroid cancer with quercetin.

**Abstract:**

Although the treatment of thyroid cancer has improved, unnecessary surgeries are performed due to a lack of specific diagnostic and prognostic markers. Therefore, the identification of novel biomarkers should be considered in the diagnosis and treatment of thyroid cancer. In this study, antibody arrays were performed using tumor and adjacent normal tissues of patients with papillary thyroid cancer, and several potential biomarkers were identified. Among the candidate proteins chosen based on the antibody array data, mature NAG-1 exhibited increased expression in tumor tissues compared to adjacent normal tissues. In contrast, pro-NAG-1 expression increased in normal tissues, as assessed by western blot analysis. Furthermore, pro-NAG-1 expression was increased when the thyroid cancer cells were treated with phytochemicals and nonsteroidal anti-inflammatory drugs in a dose-dependent manner. In particular, quercetin highly induced the expression of pro-NAG-1 but not that of mature NAG-1, with enhanced anticancer activity, including apoptosis induction and cell cycle arrest. Examination of the NAG-1 promoter activity showed that p53, C/EBPα, or C/EBPδ played a role in quercetin-induced NAG-1 expression. Overall, our study indicated that NAG-1 may serve as a novel biomarker for thyroid cancer prognosis and may be used as a therapeutic target for thyroid cancers.

## 1. Introduction

The incidence of thyroid cancer has increased in recent years, being the most commonly detected cancer in the USA (52,070 new cases in 2019) [1]. It is also the second most frequently diagnosed cancer in Korea [2]. Cancer biomarkers can be classified into three broad categories: DNA, RNA, and protein markers. New biomarkers for cancer research are highly desirable, as early detection and correct diagnosis are essential for cancer treatment. Several potential biomarkers of thyroid cancer have been reported, including *BRAFV600E* and *RAS* genes in papillary thyroid cancer (PTC) [3,4], paired box 8 (*PAX8*), peroxisome proliferator-activated receptor γ (*PPARγ*) fusion gene [5], loss of heterozygosity (LOH) on chromosome 3p and 7q loci, and *RAS* mutations in follicular thyroid cancer (FTC) [6]. However, the identification of additional biomarkers for thyroid cancer is still needed.

NAG-1 is a TGF-β superfamily cytokine with multiple roles in several diseases [7]. Two major forms of NAG-1 have been identified: a pro-form and a mature form [8]. The biological activity of mature form has been well characterized in obesity, as the mature/secreted form binds to the GFRAL receptor, leading to a reduction in appetite in the brain [9]. Additionally, the mature form of NAG-1 may play a role in pro-tumorigenic activity in some cancers [10,11]. However, the biological activity of the pro-form has not been well elucidated. The pro-forms of NAG-1 are located in the nucleus where they control the transcription of *Smad* target genes [12]. Furthermore, pro-NAG-1 alters the mitochondrial membrane potential in the cytoplasm, leading to cell death [13]. Thus, two forms of NAG-1 exhibit different activities in cancer; the pro-form shows anticancer activity, whereas the mature form shows pro-cancer activity during tumorigenesis [14]. In addition, the level of pro-NAG-1 was increased by several phytochemicals and plant extracts [15,16,17,18,19], providing a potential biomarker for anticancer compounds.

Quercetin (3,3’,4’,5,7-pentahydroxyflavone) is a flavonoid that is a major component of various plants, including raspberries, red grapes, and onions [20]. This molecule is known to have many functions, such as antioxidant, pro-apoptotic, anti-inflammatory, anti-angiogenic, and anticancer activities. Quercetin can cause cell cycle arrest and apoptosis, leading to inhibition of tumor growth, especially in breast, pancreatic, prostate, liver, and thyroid cancers [21,22,23,24,25]. Additionally, administration of sorafenib with quercetin in thyroid cancer cells can lower the dose and decrease the proliferation, adhesion, and migration properties [26]. However, the exact mechanism by which quercetin exerts this effect has not been studied, thus warranting follow-up studies.

Here, we found that the pro-form of NAG-1 is more expressed in normal thyroid tissues than in adjacent cancer tissues in papillary tumors. Furthermore, assessment of several anticancer compounds for pro-NAG-1 induction showed that quercetin is a bioactive compound that induces the expression of pro-NAG-1 but not that of mature NAG-1 in thyroid cancer cells. Moreover, quercetin induced apoptosis by inducing pro-NAG-1 expression, mediated by the transcription factor C/EBP. Our results indicate that pro-NAG-1 could be used as a useful biomarker for thyroid cancer and also provide a potential therapeutic target for the treatment of thyroid cancer with quercetin.

## 2. Materials and Methods

### 2.1. Cell Culture and Reagents

SW480, HEK293, U2OS, and BT-20 cells were purchased from the American Type Culture Collection (ATCC), whereas TPC-1, BCPAP, and HTori-3 cells were obtained from Gary Clayman (MD Anderson, Houston, TX, USA). These cells were tested by ATCC for post-freeze viability, growth properties, morphology, mycoplasma contamination, species determination (cytochrome c oxidase I assay and short tandem repeat analysis), sterility test, and human pathogenic virus testing. The cell lines were resuscitated immediately upon receiving and frozen in aliquots in liquid nitrogen. TPC-1, BCPAP, HTori-3, SW480, and BT-20 cells were cultured in Roswell Park Memorial Institute (RPMI)-1640 medium supplemented with 10% fetal bovine serum (FBS; Thermo Fisher Scientific, Waltham, MA, USA) and 1% penicillin/streptomycin (Gibco, Thermo Fisher Scientific, Waltham, MA, USA). U2OS cells were cultured in McCoy’s 5A medium supplemented with 10% FBS and 1% penicillin/streptomycin. HEK293 cells were cultured in Dulbecco’s modified Eagle’s medium (DMEM) supplemented with 10% FBS and 1% penicillin/streptomycin. All cultured cells were maintained at 37 °C under humid conditions with 5% CO_2_. Dimethyl sulfoxide (DMSO) was purchased from Biosesang (Seongnam-si, Korea), while *trans*-chalcone sulindac sulfide, piroxicam, ibuprofen, diclofenac, and acetaminophen were obtained from Sigma-Aldrich (St. Louis, MO, USA). Resveratrol was supplied by Calbiochem (San Diego, CA, USA) and tolfenamic acid by Cayman Chemical Company (Ann Arbor, MI, USA). Meloxicam, naproxen, and apigenin were purchased from Tokyo Chemical Industry (Tokyo, Japan). Quercetin, naringenin, and kaempferol were purchased from MP Biomedicals LLC (Santa Ana, CA, USA). Genistein was obtained from Acros Organics (Geel, Belgium).

### 2.2. Tissue Samples

The surgical thyroid tissues used in this study were surgical samples provided by the National Cancer Center (Goyang-si, Korea), in the form of a pair of tumors with adjacent normal tissue (Table 1). These tissue samples were collected from patients with papillary thyroid carcinoma (PTC) and subsequently stored at −80 °C. This study was approved by the Institutional Review Board of the National Cancer Center (NCC-1810150), and all the methods were conducted in accordance with the relevant guidelines and regulations.

### 2.3. Antibody Array

Whole proteins were extracted from thyroid tissues by sonication in radioimmunoprecipitation assay (RIPA) buffer (GenDEPOT, Katy, TX, USA) supplemented with proteinase and phosphatase inhibitors. The antibody array was performed using a RayBio^®^ C-Series Human Cancer Discovery Antibody Array 3 (RayBiotech, Peachtree Corners, GA, USA) according to the manufacturer’s protocol.

### 2.4. Protien Isolation and Western Blot Analysis

Cells were grown to 80% confluence and then treated with the indicated compounds. After 24 h of incubation with serum-free media, protein lysates were obtained using RIPA buffer supplemented with proteinase and phosphatase inhibitors and separated on sodium dodecyl sulfate-polyacrylamide electrophoresis (SDS-PAGE) gels (10, 12, and 15% gels for tissue samples and 12% gels for BCPAP cells). To obtain conditional media, cells were grown in a 10 cm culture dish with 10 mL of serum-free media and incubated for 24 h. The media were harvested, and the cell debris was removed and concentrated using an Amicon Ultra-15 (UFC901024; Merck Millipore Ltd., Tullagreen, Carrigtwohill, County Cork, Ireland). The separated protein bands were transferred onto nitrocellulose membranes (GVS filter technology, Zola Predosa BO, Italy) and blocked with TBS buffer containing 0.05% Tween 20 (TBS-T) with 5% non-fat milk at room temperature for 1 h, followed by overnight incubation with an appropriate primary antibody in TBS-T containing 5% non-fat milk at 4 °C. The primary antibodies used were anti-NAG-1 (specific to both pro-NAG-1 and mature NAG-1) [8], anti-galectin-3 (sc-32790; Santa Cruz Biotechnology, Santa Cruz, CA, USA), anti-osteoprotegerin (OPG; sc-390518; Santa Cruz Biotechnology, Santa Cruz, CA, USA), anti-TIMP-1(sc-6832; Santa Cruz Biotechnology, Santa Cruz, CA, USA), anti-GAPDH (sc-47724; Santa Cruz Biotechnology, Santa Cruz, CA, USA), and anti-β-actin (sc-47778; Santa Cruz Biotechnology, Santa Cruz, CA, USA). The membranes were washed three times with TBS-T buffer for 10 min and incubated with secondary antibodies dissolved in TBS-T buffer containing 5% non-fat milk at room temperature for 2 h. The membranes were washed again, and protein expression was detected by chemiluminescence using an enhanced chemiluminescence (ECL) western blotting detection reagent (Thermo Fisher Scientific, Waltham, MA, USA) on a chemiluminescence analyzer, Alliance Q9 Advanced (UVTEC CAMBRIDGE, Cambridge, England, UK) according to the manufacturer’s instructions.

### 2.5. Reverse Transcriptase Quantitative Polymerase Chain Reaction (RT-qPCR)

RNA was isolated with TRIzol reagent (Ambion, Foster City, CA, USA) according to the manufacturer’s instructions, and reverse transcription was performed using the Verso cDNA Synthesis Kit (Thermo Fisher Scientific, Waltham, MA, USA) according to the manufacturer’s instructions. RT-qPCR was performed using the QuantStudio 1 Real-Time PCR System using PowerUp™ SYBR™ Green Master Mix (A25741; Thermo Fisher Scientific, Waltham, MA, USA) with NAG-1 forward (5′-GACCCTCAGAGTTGCACTCC-3′) and reverse (5′-GCCTGGTTAGCAGGTCCTC-3′) primers. Expression of NAG-1 was corrected by GAPDH expression (forward primer: 5′-GAAGGTGAAGGTCGGAGTCA-3′, reverse primer: 5′-GACAAGCTTCCCGTTCTCAG-3′). All reactions were performed in triplicate, and the relative expression level of NAG-1 was calculated using the 2^−ΔΔCt^ method.

### 2.6. Dual-Luciferase Assay

BCPAP cells were seeded in a 24-well plate and incubated for 24 h at 37 °C. Four luciferase constructs were used (pNAG-1 -133/+70/LUC, pNAG-1 -133/+41/LUC, pNAG-1 -474/+41/LUC, and pNAG-1 -1086/+41/LUC) [27]. For the co-transfection experiments, each luciferase construct with the empty vector, p53, ATF3, CREB, RARα, C/EBPα, C/EBPδ, or EGR-1 expression vector was transfected into BCPAP cells, and either DMSO or quercetin (10 and 50 μM) was added to the transfected cells in serum-free media. After 24 h, the media were removed, and the cells were washed twice with 1x PBS. Then, 200 µL of 1x passive lysis buffer was added to each well, and the plate was shaken until the cells were detached completely on ice. The cell lysate was transferred to a new tube and centrifuged at 12,000× *g* for 15 s. Luciferase activity was measured using a Dual-Luciferase kit (Promega, Madison, WI, USA) according to the manufacturer’s protocol.

### 2.7. Cytotoxicity Assay Using High-Content Screening

Eight thousand BCPAP cells were seeded on a poly D-Lysine-coated 96-well plate and incubated overnight. Cells were treated with compounds in serum-free RPMI 1640 medium for 24 h at 37 °C. After washing with Hanks’ balanced salt solution (HBSS; Sigma-Aldrich, St. Louis, MO, USA), cells were stained with 1 μg/mL Hoechst 33342 (Sigma-Aldrich, St. Louis, MO, USA), 100 nM SYTOX^®^ Green Nucleic Acid Stain (Thermo Fisher Scientific, Waltham, MA, USA), and 25 nM MitoTracker^®^ Orange CMTMRos (Thermo Fisher Scientific, Waltham, MA, USA) in HBSS for 30 min at 37 °C. Cells were washed with HBSS twice and analyzed using the CellInsight CX7 LZR High-Content Screening (HCS) Platform (Thermo Fisher Scientific, Waltham, MA, USA) at 200× magnification.

### 2.8. Apoptosis Analysis by Flow Cytometry

BCPAP cells were cultured in a 6-well plate until they reached 60–80% confluence. The cells were treated with the compound in serum-free media and incubated for 24 h. After washing and trypsinization, cells were stained with FITC Annexin V Apoptosis Detection Kit with propidium iodide (PI; BioLegend, San Diego, CA, USA) according to the manufacturer’s protocol. The cells were then analyzed using Sony SH800 Cell Sorter (Sony Biotechnology Inc., Tokyo, Japan). The data is analyzed by FlowJo software (BD Life Sciences, Franklin Lakes, NJ, USA).2.9. Cell Cycle Analysis

Cells grown on a 6-well plate to 100% confluence were treated with the compounds in serum-free media for 24 h. Then, the cells were harvested in a microcentrifuge tube and fixed with 0.5 mL of cold 70% EtOH (Merck, Billerica, MA, USA) for 1 h. Cells were collected by centrifugation and resuspended in 0.5 mL of phosphate-buffered saline (PBS) with 0.25% Triton-X 100 (Glentham Life Sciences Ltd., Corsham, Wiltshire, UK) for 15 min on ice. After centrifugation, cell pellets were resuspended in PBS (0.5 mL) containing 10 μg/mL RNase A (iNtRON Biotechnology, Seongnam, Gyeonggi Province, Korea) and 20 μg/mL PI (Invitrogen, Carlsbad, CA, USA) and incubated for 30 min in the dark. Cells were analyzed by Sony SH800 Cell Sorter (Sony Biotechnology Inc., Tokyo, Japan). The data is analyzed by FlowJo software (BD Life Sciences, NJ, USA).2.10. Transient Transfection of NAG-1 Construct

BCPAP cells were seeded in 60 mm dishes and transiently transfected with PolyJet Transfection Reagent (SignaGen, Gaithersburg, MD, USA) according to the manufacturer’s protocol. After 24 h post-transfection, serum-free media containing DMSO and quercetin (1, 10, and 50 μM) was added to the dishes. After 24 h of treatment with the compounds, the cells were harvested and analyzed by western blotting.

### 2.9. Enzyme-Linked Immunosorbent Assay (ELISA)

Plasma NAG-1 levels were measured using the Human GDF15 Quantikine ELISA Kit (DGD150; R&D Systems, Minneapolis, MN, USA). Samples, reagents, and buffers were prepared according to the manufacturer’s instructions. The detection sensitivity of NAG-1 was 4.39 pg/mL, and the assay range was 23.4–1500 pg/mL. To determine the optical density, a microplate reader was used to measure the intensity of the wells. The microplate reader was set to 450 nm and corrected by subtracting the intensity at 570 nm. The concentration of each sample was calculated using a standard curve.

### 2.10. Statistical Analysis

Statistical analysis was conducted using Microsoft Office Excel, SPSS, and GraphPad Prism 8. Unpaired Student’s *t*-test and one-way analysis of variance were used to analyze the data. For all analyses, results were considered significant at *p* < 0.05 (* *p* < 0.05, ** *p* < 0.01, and *** *p* < 0.001).

## 3. Results

### 3.1. Identification of Differentially Expressed Proteins in Thyroid Normal and Tumor Tissues

To identify novel biomarkers in thyroid cancer, we performed an antibody array using normal human thyroid and tumor tissues. Tumor and adjacent normal tissues were obtained from patients with papillary thyroid cancer, and an antibody array was performed. Four candidate proteins, galectin-3, NAG-1, TIMP-1, and osteoprotegerin (OPG), were identified as potential biomarkers of thyroid cancer (Figure 1A). The protein expression of galectin-3, TIMP-1, and NAG-1 was higher in tumor tissues than in normal tissues. In contrast, OPG expression was higher in normal tissues than in tumor tissues. To confirm the results of the antibody array, western blot analysis was performed using the proteins extracted from three patients with papillary cancer (Table 1). The expression of OPG was higher in normal tissues, whereas that of galectin-3 and TIMP-1 was higher in tumor tissues (Figure 1B). Interestingly, two forms of NAG-1/GDF15 were detected in the tissues. Mature NAG-1 was expressed more in tumor tissues, whereas pro-NAG-1 exhibited higher expression in normal tissues. Since NAG-1 is expressed as a pro- (~35 kDa) and a mature form (~12 kDa), the antibody array data were consistent with the mature form of NAG-1. Pro-NAG-1 and mature-NAG-1 have been reported to exhibit different biological activities in tumorigenesis [7,14]. Since mature NAG-1 has been detected in the antibody array and mature serum NAG-1 is linked to thyroid pro-tumorigenesis [28], we measured serum NAG-1 levels to identify the linkage in different types of thyroid cancer: non-aggressive benign thyroid nodules (BTN: FA and NH) and aggressive differentiated thyroid cancer (DTC: FVPTC, PTC, and FTC). Serums from forty-nine patients (Appendix A) were obtained, and NAG-1 expression was measured by ELISA. The data were analyzed for NAG-1 concentration by tumor type, sex, BMI, and age. NAG-1 concentration was lower in the plasma samples of patients with benign tumors, such as follicular adenoma and nodular hyperplasia than in those with malignant thyroid cancer, such as PTC, FVPTC, and FTC, but the difference was not statistically significant (Appendix A). In addition, no difference was noted in NAG-1 concentrations according to sex and BMI (Appendix A). Interestingly, the concentration of NAG-1 in the plasma increased with patient age [29] (Appendix A). According to these data, the levels of mature serum NAG-1 change with age regardless of the patient’s sex, BMI, or cancer type. Together, these results indicate that pro- and mature NAG-1 not only exhibit great potential as a biomarker for diagnosis but also as a therapeutic target for thyroid cancer.

### 3.2. Mature and Pro-NAG-1 Expression in Various Cancer Cells

To identify whether various cancer cells express NAG-1 at the transcriptional level, we first measured the mRNA levels of NAG-1 in thyroid and other cancer cells. As shown in Figure 2A, BCPAP cells exhibited higher expression of NAG-1 mRNA among thyroid cancer cell lines, whereas U2OS osteosarcoma cells showed the highest NAG-1 mRNA expression. Since NAG-1 is synthesized as pro-NAG-1 and cleaved into mature NAG-1, western blot analysis was conducted to determine the mature and pro-NAG-1 expression in various cancer cell lines. Western blot analysis revealed that BCPAP cells showed higher NAG-1 expression in the cell lysates and conditioned medium among the thyroid cancer cells (Figure 2B), which was consistent with the RT-qPCR data. Among the non-thyroid cancer cells, BT-20 breast cancer cells and U2OS osteosarcoma cells expressed significant amounts of NAG-1 in cell lysates and conditioned media (Figure 2C).

### 3.3. Quercetin Increases Pro-NAG-1 Levels but Not Mature NAG-1 Levels

Since two forms of NAG-1 (mature and pro-) are differentially expressed in the cell lysates with opposing activities in cancer cells, we examined several compounds that increase pro-NAG-1 expression in thyroid cancer cells. BCPAP cells were treated with various anticancer compounds, such as phytochemicals and nonsteroidal anti-inflammatory drugs (NSAIDs). Among these, sulindac sulfide and quercetin dramatically increased pro-NAG-1 expression compared to DMSO treatment (Figure 3A). Treatment with sulindac sulfide (SS), *trans*-chalcone (TC), and quercetin (QUE) also increased pro-NAG-1 expression in a dose-dependent manner (Figure 3B). Since quercetin treatment increased pro-NAG-1 expression among the tested compounds, cells were treated with different doses of quercetin, and pro- and mature NAG-1 expressions were measured. Interestingly, quercetin only increased pro-NAG-1 but did not alter the expression of mature NAG-1 in a dose-dependent manner (Figure 3C). This result indicates that pro-NAG-1 expression is preferentially increased by quercetin, and pro-NAG-1 is a chemotherapeutic target for thyroid cancer.

### 3.4. Quercetin Induces Apoptosis and Cell Cycle Arrest

To confirm the anticancer activity of quercetin, a high-throughput platform was used. Representative images showed reduced fluorescence intensity of SYTOX and MitoTracker in quercetin-treated cells (Figure 4A). Cell permeability was increased (Figure 4B), and mitochondrial membrane potential was decreased by quercetin treatment in a dose-dependent manner (Figure 4C), indicating apoptosis induction. Annexin V assay was performed to measure the percentage of apoptotic cells in quercetin-treated cells. The data showed that quercetin increased the percentage of apoptotic cells (Figure 4D). Additionally, PI staining data suggested that quercetin treatment affected cell cycle arrest at the S phase (Figure 4E). Taken together, quercetin induced cell apoptosis and cell cycle arrest.

### 3.5. Quercetin Increases NAG-1 Promoter Activity through p53, C/EBPα, and C/EBPδ

To determine the mechanism by which quercetin affects NAG-1 expression at the transcriptional level, we examined NAG-1 promoter activity in the presence of quercetin. First, quercetin induced NAG-1 mRNA expression in a dose-dependent manner (Figure 5A). To elucidate the molecular mechanism, we conducted a dual-luciferase assay using several NAG-1 promoters linked to the luciferase gene. Several luciferase constructs, including pNAG-1-133/+70/LUC, pNAG-1-133/+41/LUC, pNAG-1-474/+41/LUC, pNAG-1-1086/+41/LUC, and pRL null construct, were co-transfected into BCPAP, and transfected cells were treated with DMSO or 50 μM quercetin. After 24 h of incubation, luciferase activity was measured, and NAG-1 promoter activity was marginally increased by quercetin in all constructs (Figure 5B), indicating that the quercetin response element may be located within the −133 to +70 region. Furthermore, this promoter region was examined in the presence of quercetin, revealing that quercetin increased NAG-1 promoter activity in a dose-dependent manner (Figure 5C). Quercetin increases the level of p53 tumor suppressor protein in human colorectal cancer cells [30]. To investigate whether the increase in NAG-1 promoter activity depends on p53, we transfected an empty vector or p53 expression vector with pNAG-1-133/+70/LUC and pRL null and treated them with DMSO or quercetin. The luciferase activity was higher in the p53-transfected group (Figure 5D), indicating that quercetin increased NAG-1 promoter activity via p53 expression. However, since BCPAP is a p53 mutant cell line, we hypothesized that there would be another pathway. To identify additional factors that cause NAG-1 induction, we used the pNAG-1-133/+41/LUC construct to perform the dual-luciferase assay. ATF3, CREB, RAR, C/EBPα, C/EBPδ, or EGR-1 can bind to the NAG-1 promoter within the -133 bp region [15,31,32]. Thus, several expression vector plasmids were co-transfected with the NAG-1 promoter in BCPAP cells. As a result, C/EBPα and C/EBPδ levels were significantly increased compared to the empty vector (EV)-transfected group (Figure 5E), in a dose-dependent manner (Figure 5F). Taken together, quercetin may affect C/EBPα and C/EBPδ expression, followed by the induction of NAG-1 expression in BCPAP cells.

## 4. Discussion

Although thyroid cancer is a common and relatively indolent cancer with a low mortality rate, some types of thyroid cancer show aggressive clinical features, such as rapid progression, lymph node, distant metastases, and even death from persistent and recurrent disease. Therefore, it is important to identify reliable and clinically applicable novel biomarkers for thyroid cancer diagnosis and prediction.

To elucidate this problem, we conducted an antibody array using papillary thyroid cancer and normal tissues. Four proteins were identified that showed increasing or decreasing expression in thyroid tumorigenesis (Figure 1A,B). Galectin-3 controls cellular proliferation and apoptosis in normal cells as well as malignant transformation and metastasis in cancer cells [33]. Galectin-3 is a notable protein marker for thyroid tumors, and we confirmed the induction of galectin-3 in an antibody array and western blot analysis (Figure 1A,B). The expression of TIMP-1 in the plasma and tissues of patients with cancer is highly increased, with more significant levels related to worse clinical results in various cancers, including prostate and colon cancers [34]. However, it is not clear whether TIMP-1 serves only as a biomarker of cancer progression or functions to promote cancer progression; thus, it could serve as an important cancer therapeutic target in thyroid cancer. Osteoprotegerin, which is engaged in many biological systems, plays a key role in the regulation of bone resorption [35]. The use of serum OPG as a prognostic marker has also been investigated in breast cancer and was found to be a potential diagnostic marker [36]. However, OPG was increased in normal thyroid tissues according to antibody array and western blot data, highlighting the differences between tissue and serum OPG levels. The exact biological activity of OPG in thyroid cancer remains to be elucidated.

NAG-1/GDF15 has been identified as an NSAID-induced gene [8]. It is by several anti-cancer agents, including phytochemicals [37], NSAIDs [27], and PPARγ ligands [38]. Although the biological activity of NAG-1 in obesity has been well established [39], the role of NAG-1 in tumorigenesis is contradictory in several cancers [7]. In general, an anti-tumorigenic effect during tumor development was observed in transgenic mice expressing NAG-1 [40,41]. In contrast, most results showing the pro-cancer activity of NAG-1 were obtained from in vitro experiments using cultured cells [42]. Recently, Kang et al. reported that NAG-1/GDF15 is a mitokine that increases the invasiveness of thyroid cancer [43]. This discrepancy may result from the different activities of pro-NAG-1 and mature NAG-1, the different expression of mature and pro-NAG-1, or multiple activities of NAG-1 depending on the cell context. In this study, antibody array data indicated that NAG-1 expression was increased in tumor tissues due to the abundant expression of mature NAG-1. However, size differentiation by western blot analysis clearly indicated that pro-NAG-1 was more highly expressed in normal tissues, whereas mature NAG-1 was more highly expressed in tumor tissues. This is consistent with our previous report that NAG-1 may function as a moonlighting protein in tumorigenesis [12,14].

Many pharmacological approaches have been proposed for conventional drug therapies. In the present study, we screened dietary compounds, phytochemicals, and NSAIDs to examine their effects on pro-NAG-1 induction in thyroid cancer cells. The levels of pro-NAG-1 were increased by sulindac sulfide or quercetin, a conventional NSAID and a phytochemical, respectively; these have been reported to increase pro-NAG-1 levels and exert anticancer activity in several cancers [44,45]. Interestingly, in the presence of quercetin, only pro-NAG-1 expression was increased in BCPAP cells (Figure 3C). This result supports the fact that many phytochemicals induce anticancer activities via NAG-1 expression, even though mature NAG-1 is linked to pro-tumorigenic activity. Although detailed mechanisms need to be elucidated, this is the first report to suggest that a phytochemical preferentially increases the expression of pro-NAG-1 but not that of mature NAG-1 in BCPAP cells.

Several transcriptional factors have been shown to increase NAG-1 expression at the transcriptional level. Among these, C/EBPα and C/EBPδ were identified as NAG-1 inducers at the transcriptional level. The overexpression of CCAAT/enhancer-binding protein (C/EBP) α, β, and δ caused a significant increase in basal and capsaicin-induced NAG-1 promoter activity [46], and quercetin increased C/EBPβ mRNA and protein expression [47]. Thus, it is likely that quercetin increases the expression of C/EBP isotypes, followed by increased NAG-1 expression. However, the detailed molecular mechanism by which quercetin affects NAG-1 expression at the transcriptional level remains to be elucidated.

Taken together, this study highlights a potential biomarker for the diagnosis of thyroid cancer, especially in differentiating between pro- and mature NAG-1. Further investigation may be required to elucidate the molecular mechanism of quercetin-induced NAG-1 expression; however, our data indicate that C/EBP proteins contribute at least, in part, to the quercetin-induced NAG-1 expression (Figure 6).

## 5. Conclusions

The expression of mature NAG-1 in BCPAP cells was not significantly altered in the presence of quercetin, but pro-NAG-1 expression was significantly higher. This report suggests that pro-NAG-1 may be used as a therapeutic target in thyroid cancer.

## Figures and Tables

**Figure 1 cancers-13-03022-f001:**
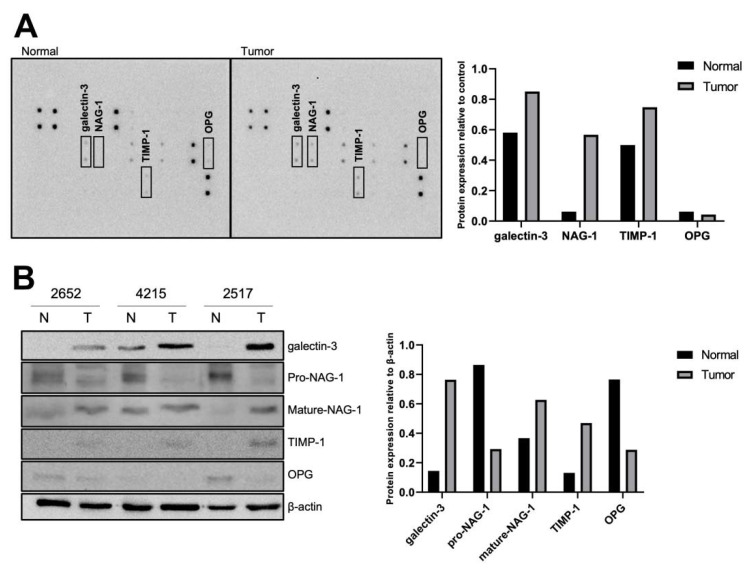
Antibody array using human thyroid tissues. (**A**) Antibody array (RayBio^®^ C-Series Human Cancer Discovery Antibody Array 3, RayBiotech) showed that 30 cytokines related to cancer biology were differentially expressed between thyroid normal and tumor tissues. One pair of PTC samples was used to analyze the expression of these cytokines. Four different proteins (Galectin-3, NAG-1/GDF15, TIMP-1, and osteoprotegerin) were selected as biomarker candidates for the diagnosis of thyroid cancer (rectangle). The right graph represents the intensity of each protein. (**B**) Western blot was performed using three pairs of PTC samples to confirm the data of antibody array. The protein expression of galectin-3, mature NAG-1, pro-NAG-1, TIMP-1, and OPG was examined using thyroid tissue samples. The right graph is from the average quantification of protein expression from three patients. N, thyroid normal tissue; T, thyroid tumor tissue. The number of patients is indicated (see Table 1 for details). Uncropped versions of blots presented in Appendix A.

**Figure 2 cancers-13-03022-f002:**
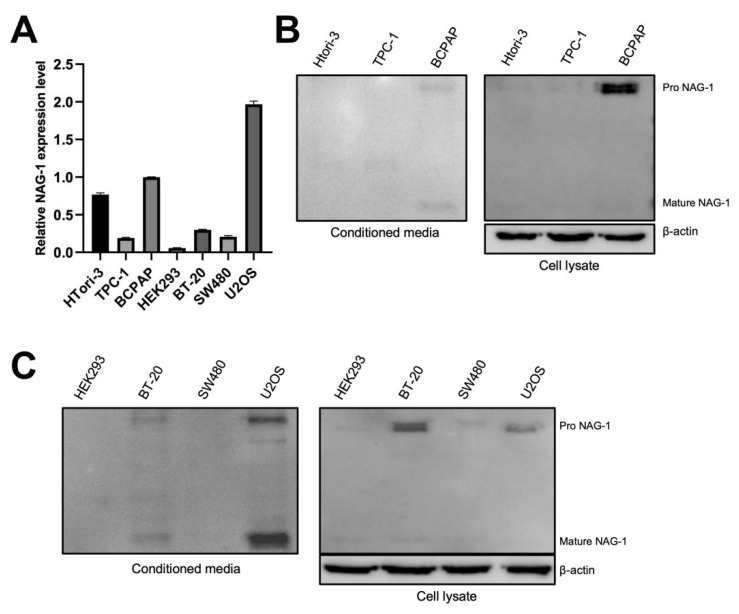
NAG-1 expression in various types of cancer cell lines. (**A**) Quantitative reverse transcriptase-polymerase chain reaction (RT-qPCR) was conducted using both thyroid (HTori-3, TPC-1, and BCPAP) and non-thyroid cell lines (HEK-293, BT-20, SW480, and U2OS). Analysis was performed in triplicate, and the graph shows the mRNA levels of NAG-1 expression compared with those of BCPAP cells. Western blot analysis using (**B**) thyroid and (**C**) non-thyroid cancer cells. Conditioned medium and total cell lysates were isolated as described in the Methods section, and NAG-1 antibodies against pro- and mature NAG-1 were hybridized into the membrane. Uncropped versions of blots presented in the Appendix A.

**Figure 3 cancers-13-03022-f003:**
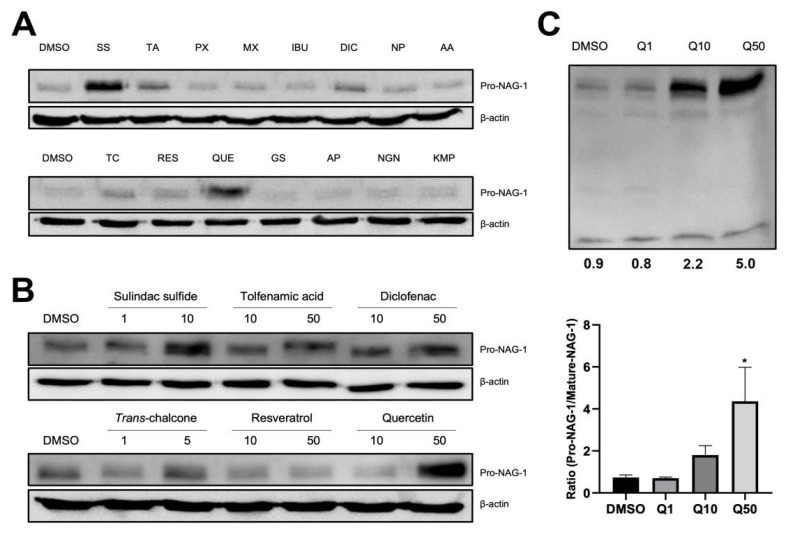
Pro-NAG-1/GDF15 expression in the presence of several anticancer compounds. (**A**) The human thyroid cancer cell line BCPAP was exposed to different nonsteroidal anti-inflammatory drugs (NSAIDs) and phytochemicals for 24 h with serum-free media (sulindac sulfide (SS): 10 μM, *trans*-chalcone (TC): 5 μM; tolfenamic acid (TA), piroxicam (PX), meloxicam (MX), ibuprofen (IBU), diclofenac (DIC), naproxen (NP), acetaminophen (AA), resveratrol (RES), quercetin (QUE), genistein (GS), apigenin (AP), naringenin (NGN), and kaempferol (KMP): 50 μM). Cells were harvested, and the proteins were extracted with radioimmunoprecipitation assay (RIPA) buffer. Proteins were subjected to western blot to detect pro-NAG-1 and β-actin expression. (**B**) BCPAP cells were exposed to different doses (sulindac sulfide; 1, 10 μM, *trans*-chalcone: 1, 5 μM; others: 10, 50 μM) of anticancer compounds for 24 h with serum-free media. Western blot analysis was conducted as described in the Methods section. (**C**) Quercetin at doses of 1, 10, and 50 μM were treated to BCPAP cells. Pro- and mature NAG-1 were detected by western blot. The bottom number represents the ratio of pro- and mature NAG-1 expression, three independent experiments and a statistical analysis were done. * *p* < 0.05. Uncropped versions of blots presented in the Appendix A.

**Figure 4 cancers-13-03022-f004:**
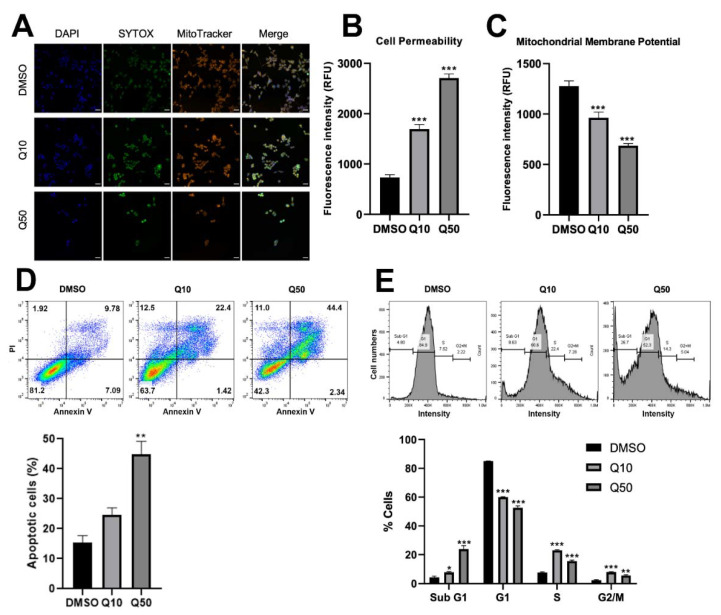
Quercetin induced apoptosis and cell cycle arrest in BCPAP cells. (**A**) BCPAP cells were subjected to high-content screening using the CellInsight CX7 LZR High-Content Screening (HCS) Platform (Thermo Fisher Scientific, Waltham, MA, USA), and representative images are shown. Cells were stained with 1 μg/mL Hoechst 33342, 100 nM SYTOX^®^ Green Nucleic Acid Stain (green), and 25 nM MitoTracker^®^ Orange CMTMRos (orange) in Hanks’ Balanced Salt Solution (HBSS) for 30 min. DMSO, vehicle; Q10, 10 μM quercetin; Q50, 50 μM quercetin. Magnification = 200×, scale bar = 200 μm. CellInsight CX7 LZR High-Content Screening (HCS) Platform data were quantified and presented as changes in cell permeability (**B**) and mitochondrial membrane potential (**C**) as measured by the average intensities of SYTOX and MitoTracker. Values are expressed as the mean ± the standard error of the mean (SEM). *** *p* < 0.001.(**D**) Apoptosis analysis using Annexin V/ propidium iodide (PI) staining. DMSO, vehicle; Q10, 10 μM quercetin; Q50, 50 μM quercetin. The bottom graph represents the percentage of cells in the early and late apoptotic populations. Results are representative of three independent experiments. ** *p* < 0.01. (**E**) Cell cycle analysis using PI. The representative graph is shown at the top. The bottom graph indicates the different cell cycle populations in the presence of quercetin. Results are representative of three independent experiments. * *p* < 0.05, ** *p* < 0.01, and *** *p* < 0.001.

**Figure 5 cancers-13-03022-f005:**
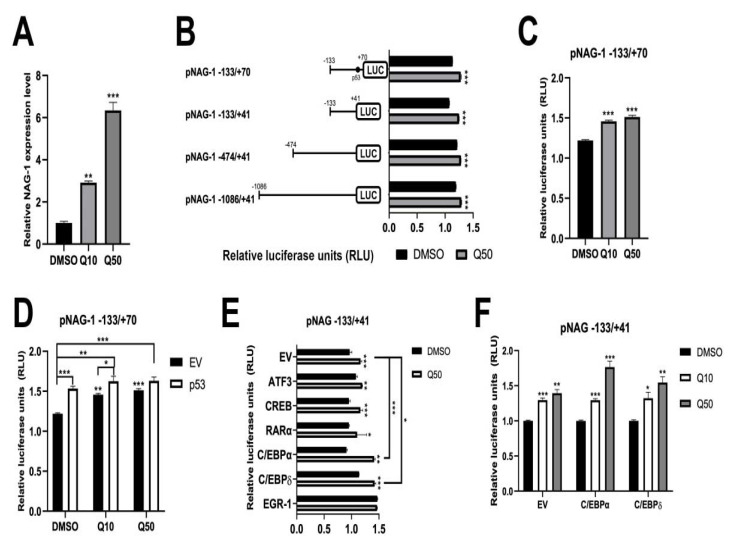
Quercetin increased NAG-1 promoter activity. (**A**) Quercetin increased NAG-1 expression at the transcriptional level. GAPDH was used as a control. DMSO, vehicle; Q10, 10 μM quercetin; Q50, 50 μM quercetin. Data are representative of three independent samples. ** *p* < 0.01 and *** *p* < 0.001. (**B**) The pNAG-133/+70/LUC, pNAG-133/+41/LUC, pNAG-474/+41/LUC, pNAG-1086/+41/LUC, and pRL null constructs were co-transfected into BCPAP cells, and cells were treated with dimethyl sulfoxide (DMSO) or Q50. Luciferase activity was measured using the Dual-Luciferase system. The X-axis represents relative luciferase units, and the schematic diagram of the NAG-1 promoter is indicated on the left of the figure. *** *p* < 0.001. (**C**) DMSO or quercetin (10 and 50 μM) was treated in pNAG-1-133/+70/LUC containing the p53-binding site and pRL null-transfected BCPAP cells. *** *p* < 0.001. (**D**) BCPAP cells were co-transfected with pNAG-1-133/+70/LUC and pRL null constructs and then divided into two groups. One group was transfected with the empty vector (EV) and another group was transfected with the p53 expression vector. After 24 h of incubation, the transfected cells were treated with DMSO or quercetin for 24 h. * *p* < 0.05, ** *p* < 0.01, and *** *p* < 0.001. (E) BCPAP cells were co-transfected with pNAG-1-133/+41/LUC and pRL null constructs and then divided into two groups. One group was transfected with the empty vector (EV) and another group was transfected with ATF3, CREB, RARα, C/EBPα, C/EBPδ, or EGR-1 expression vector. After 24 h of stabilization, the transfected cells were treated with DMSO or quercetin. Data are representative of three independent samples. * *p* < 0.05, ** *p* < 0.01, and *** *p* < 0.001. (**F**) pNAG-1 -133/+41/LUC- and pRL null-transfected BCPAP cells were treated with DMSO or quercetin (10 and 50 μM). Data are representative of three independent samples. All the luciferase experiments were conducted and represented as * *p* < 0.05, ** *p* < 0.01, and *** *p* < 0.001.

**Figure 6 cancers-13-03022-f006:**
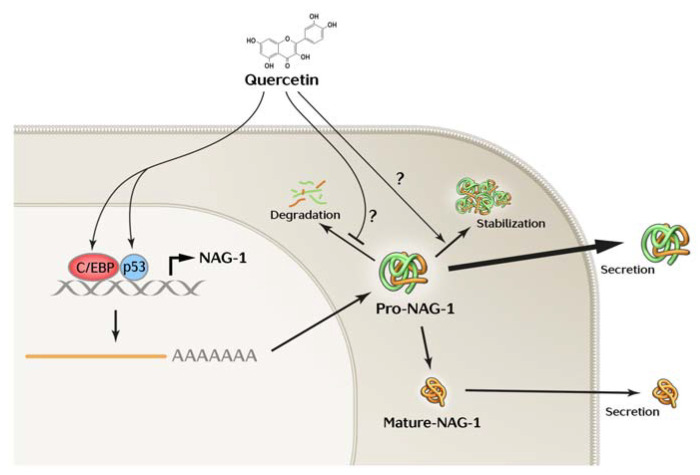
Proposed mechanism by which pro-NAG-1 plays an anti-tumorigenic role in thyroid cancer. Pro-NAG-1 and mature NAG-1 are produced in thyroid tissues and cancer cells. Quercetin Appendix A in human thyroid cancer, leading to an increase in pro-NAG-1 only, and not mature NAG-1.

**Table 1 cancers-13-03022-t001:** Patient sample information used in this study.

Patient Number	Gender	Age at Time of Surgery	Type of Thyroid Cancer
2652	M	35	Papillary carcinoma
4215	F	56	Papillary microcarcinoma
2517	M	34	Papillary carcinoma

## Data Availability

Not applicable.

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
