# Peer review of "Quercetin Induces Anticancer Activity by Upregulating Pro-NAG-1/GDF15 in Differentiated Thyroid Cancer Cells"

_cancers, 2021, doi:10.3390/cancers13123022_

Round 1

Reviewer 1 Report

Manuscript entitled „Quercetin induces anticancer activity by upregulating pro-2 NAG-1/GDF15 in differentiated thyroid cancer cells” by Yukyung Hong et al. discusses one of the important issues of searching for new biomarker and potential therapeutic target for the treatment of thyroid cancer. The authors have proposed that NAG-1 may serve as a novel biomarker for thyroid cancer prognosis and may be used as a therapeutic target for thyroid cancers. However, despite the interesting topic touched upon in the manuscript, I figure that the manuscript is not suitable for publication in current form in Cancers. I have not been able to carry out a thorough revision of the present manuscript due to the illegible form of the figures included. Their resolution does not allow verification of the results described. Therefore, I believe it is necessary to re-review the manuscript after improving the quality of the figures.

Author Response

Reviewer 1
Manuscript entitled „Quercetin induces anticancer activity by upregulating pro-2 NAG-1/GDF15 in differentiated thyroid cancer cells” by Yukyung Hong et al. discusses one of the important issues of searching for new biomarker and potential therapeutic target for the treatment of thyroid cancer. The authors have proposed that NAG-1 may serve as a novel biomarker for thyroid cancer prognosis and may be used as a therapeutic target for thyroid cancers. However, despite the interesting topic touched upon in the manuscript, I figure that the manuscript is not suitable for publication in current form in Cancers. I have not been able to carry out a thorough revision of the present manuscript due to the illegible form of the figures included. Their resolution does not allow verification of the results described. Therefore, I believe it is necessary to re-review the manuscript after improving the quality of the figures.

Response) As the reviewer suggested, we changed all the figures with high-resolution figures in the text.

Reviewer 2 Report

This very nicely designed and written project.

My humble suggestion would be clarified the diagnostic criteria (based on which edition of WHO) of the human tissue samples. If there are subtype information for PTC. And while saying FV-PTC, if they are infiltrative type or invasive type. Of those FV-PTC diagnosed cases, NIFTP were excluded? Were there any oncocytic morphology? Oncocytic morphology or Hürthle cell tumours were excluded?

Author Response

Reviewer 2

My humble suggestion would be clarified the diagnostic criteria (based on which edition of WHO) of the human tissue samples.

Response) Our diagnosis is based on the latest WHO classification which was published in 2017.

If there are subtype information for PTC. And while saying FV-PTC, if they are infiltrative type or invasive type. Of those FV-PTC diagnosed cases, NIFTP were excluded?

Response) We diagnosed follicular variant papillary thyroid carcinoma with two main subtypes: infiltrative subtype and encapsulated with invasion type. In addition, we diagnosed NIFTP (not FV-PTC) after thorough examination of tumor capsule with absence of invasion.

Were there any oncocytic morphology? Oncocytic morphology or Hürthle cell tumours were excluded?

Response) Inthis study, pure form of oncocytic variant PTC is distinguished from other variants of PTC.

Round 2

Reviewer 1 Report

Manuscript entitled „Quercetin induces anticancer activity by upregulating pro-2 NAG-1/GDF15 in differentiated thyroid cancer cells” by Yukyung Hong et al. presents interesting research and fits into the theme of the journal „Cancers”. However, the authors should correct and clarify a number of points before publishing.

  • Quercetin Induces Apoptosis and Cell Cycle Arrest - Cell cycle analysis is wrong. Aggregates and doublets must be removed before the analysis of cell cycle. This section must be re-analysed. To obtain a reliable cell cycle profile you should use appropriate software and algorithms (i.e. FlowJo, ModFit, FSCExpress). You can try to download the 30 days free version of FCSExpress that have the cell cycle analysis software.
  • Line 88 and 90 - no manufacturers McCoy’s and DMEM medium
  • In the pdf file received for review, the quality of the figures is lower than the original. I do not know whether the problem lies with the authors who pasted the figures into the template or with the editor. This applies especially to figure 4A. This should be corrected.
